# Mild Winter Causes Increased Mortality in the Fall Webworm *Hyphantria cunea* (Lepidoptera: Arctiidae)

**DOI:** 10.3390/insects14060534

**Published:** 2023-06-07

**Authors:** Takahiro Matsuura, Rohit Bangay, Nobuko Tuno

**Affiliations:** School of Natural Sciences and Technology, Kanazawa University, Kanazawa 920-1192, Japan

**Keywords:** global warming, winter survival, insect population, insect extinction

## Abstract

**Simple Summary:**

There is concern that global warming will change and expand the habitat of insect pests. In 1945, the fall webworm *Hyphantria cunea* (Drury) was introduced from North America to Japan, and for several decades it had been growing in large numbers on roadside trees and garden trees in various parts of Japan. However, it has disappeared from where it used to have outbreaks and is now found only in areas with high latitudes or high altitudes. In Kanazawa City, which is in the middle of Japan, 18 years of weekly data on the occurrence of this species are recorded. We collated records of the occurrence of this species with annual meteorological data. We found that the number of individuals in the generation that entered dormancy in the fall and emerged in the following spring was negatively correlated with the warmth of the winter. When this phenomenon was manipulated in the laboratory, pupae stored under warm winter conditions showed significant weight loss and a high mortality rate. Many of the pupae that died were infected with fungi. These results suggest that the populations of *Hy. cunea* have disappeared in warmer southwestern Japan due to the mild winter, and currently survive only in the high-altitude areas of Japan’s northern and central mountainous regions. This result further predicts that the distribution of this species, which is currently invading the world, will be strongly affected by global warming and will continue to change in the future.

**Abstract:**

The fall webworm *Hyphantria cunea* (Drury) is native to North America and Mexico and has currently expanded its distribution to the temperate areas of the Northern Hemisphere including Japan. According to the data on seasonal fluctuations of this moth for 18 years collected in western-central Japan, the abundance of adults of the overwintered generation showed a negative correlation with winter temperature. We investigated survival, weight loss, and fungal infection of diapausing pupae at 3.0 (an approximate temperature of cold winter) and 7.4 °C (a temperature of mild winter). In the results, mortality was higher and weight loss was larger in pupae exposed to 7.4 °C than in those exposed to 3.0 °C. In addition, pupae that were heavier at the start of cold exposure survived longer than lighter ones. Furthermore, almost all pupae that died at 7.4 °C were infected by fungi. It has been reported that the distribution range of this moth shifts to higher latitudes. According to the experiments conducted, it has been observed that warm winters can lead to a decrease in pupae weight and an increase in fungal deaths; however, the impact of warm winters on populations in the field can be more complicated and multifaceted.

## 1. Introduction

Insect population persistence and growth are seriously affected by microclimate and global climate changes they experience. The ICPP assessment sixth report (2022) predicts that invertebrates (15%), amphibians (11%), and flowering plants (10%) are at high risk of extinction for mid-levels of projected warming (3.2 °C). On the other hand, the spread and establishment of alien species attributed to climate change have increased, helped by longer seasons and warmer weather [1]. For example, the Asian tiger mosquito is now prevalent in several southern European countries but projections show it is likely to extend its range further north [2]. In terms of global warming, the expansion of biological distribution tends to attract attention but shrinks in distribution have also been reported [3,4]. We need to carefully explore species-specific information to predict insect species population growth or decline by global warming in the changing world.

The fall webworm *Hyphantria cunea*, a polyphagous folivore pest that originated in North America and Mexico, currently occurs throughout temperate areas of the Northern Hemisphere [5]. In Japan, it was first recorded in 1945 from Tokyo and has rapidly expanded its distribution [6]. All Japanese populations were bivoltine (i.e., they pass two generations in a year) by the early 1970s but trivoltine individuals have increased since the late 1970s in southern Japan [6,7]. According to Masaki (1975) [6], its life history in the bivoltine areas is as follows. Adult moths emerge from overwintered pupae in April and May, soon lay eggs, and die. Their first-generation progenies grow from May to June, emerge as adults from early to midsummer, soon lay eggs, and die. Second-generation progenies grow from midsummer to early autumn and enter diapause at the pupal stage for overwintering.

To develop a control program for this pest, the city government of Kanazawa in western-central Japan has monitored its seasonal fluctuations since 2004 using pheromone traps (Nittler type, Idemitsu Agri Co., Ltd., Tokyo, Japan); traps were set at nine points in the city during entire active seasons (i.e., from late April to early October) and the number of males caught was recorded weekly. Our initial goal was to discover a straightforward way to forecast the moth population for the upcoming year using meteorological data. According to our preliminary analysis of these monitoring data, the number of catches in spring decreased when winter temperatures were higher on average, suggesting that overwintering mortality is high in mild winter. Aoki [8] and Takezawa et al. [9] reported that the major mortality factors of this species during winter diapause were parasitism by flies (approximately 28%) and wasps (20%) and infection by filamentous fungi (15%). Among these parasites, fungi may show different responses to temperature from insect species; most fungi are able to proliferate even at temperatures below 10 °C [10], whereas larvae of this moth are unable to grow below 10 °C [11]. Therefore, a fungal infection could have a profound effect on pupal mortality in mid-winter when the temperature usually falls below 10 °C. In Kanazawa, the January and February mean temperatures in a normal year are 3.8 and 3.9 °C, respectively, but were 5.1 and 6.2 °C in 2007 when the spring catch extremely decreased in the city. It is, therefore, possible that the higher performance of fungi relative to that of this moth in mild winter has caused higher mortality in this moth. To test this hypothesis, we investigated whether temperature during winter diapause affects mortality and fungal infection in this species. In addition, a DNA barcoding study was carried out to identify the fungal species isolated from the overwintering pupae.

## 2. Materials and Methods

### 2.1. Effects of Climatic Conditions on the Abundance of Overwintered Adult Individuals

#### Insect Monitoring

The population fluctuation of the fall webworm was monitored during its active season (from late April to early October) at nine points in Kanazawa City, Ishikawa Prefecture (36°56′ N, 136°66′ E, 15–35 m in altitude), using pheromone traps (Nitolure, Idemitsuagri Co., Ltd., Tokyo, Japan) for 18 years from 2004 to 2021. Figure 1 shows the average number of weekly captures of males for 18 years. The abundance clearly has two peaks in a year, the first in the fifth week after the start of collection and the second in the fourteenth week. The first peak was assumed to be composed of individuals that emerged from overwintered pupae, while the second peak would be composed of individuals of the next generation. In this study, individuals collected from the first to the tenth week of each year were summed and assigned as a surrogate of the abundance of adult individuals of the overwintered generation (N1) in that year.

### 2.2. Climatic Factors

A number of climatic factors are considered to affect the abundance of the overwintered population of this moth. For example, if winter is cold, the survival of overwintering pupae may be reduced. If spring is cold, i.e., if daily temperatures rarely exceed 10 °C (“developmental threshold” below which the development of this moth does not proceed [11]) the emergence of adult moths may be retarded. Furthermore, if the temperature in spring is high, their emergence may also be inhibited because the development of this moth was retarded at temperatures above 25 °C [12,13]. Here, we analyzed the effects of winter coldness and temperature conditions in spring to early summer (April to June) on the abundance of adult individuals of the overwintered generation (N1) by GAM. In this study, the mean temperature from January to February was used as a surrogate of winter coldness. On the other hand, daily mean temperatures above 10 °C were cumulated (Σ (daily mean temperature −10)) in April, May, and June if it exceeded 10 °C and assigned as “cumulative effective temperature for growth” in the corresponding month. For a representative of “heat sum” in June, daily temperatures above 25 °C were cumulated in June if daily temperature exceeded 25 °C as above (daily mean temperature in April and May rarely exceeded 25 °C). The temperature data in Kanazawa were obtained from the Meteorological Agency of Japan (https://www.data.jma.go.jp/gmd/risk/obsdl/index.php, accessed on 1 June 2022). Then, GAM was performed with the N1 as a response variable and winter coldness and temperature conditions in spring as explanatory variables.

### 2.3. Effects of Cold Temperatures on Pupal Mortality

#### 2.3.1. Insects and Rearing

*Hyphantria cunea* larvae are gregarious and form net nests when they are young. We collected young (1st to 2nd instar) larvae from various locations in Kanazawa City in late July and early August 2017. According to the monitoring data presented in Figure 1, they were destined to enter pupal diapause and overwinter. Collected larvae were divided into two diet groups and raised with either mulberry (*Morus bombycis*) or sweetgum (*Liquidambar styraciflua*) leaves. Rearing was carried out with mesh cages (20 cm × 20 cm × 30 cm) under a diapause-inducing condition, a constant temperature of 25 °C, and a photoperiod of 14-h-light:10-h-dark [7]. Every day, larvae were supplied with fresh leaves that were collected from trees grown on the Kakuma campus of Kanazawa University (ca. 100 m in altitude). When they pupated, they were individually weighed and kept in glass tubes (10 mL in volume) loosely closed with autoclaved caps for ventilation. These tubes and caps were weighed before use, and therefore pupal weight can be determined without taking out the pupae from tubes. This procedure was to avoid unintentional injury at the time of weighing.

#### 2.3.2. Survival, Fungal Infection, and Weight Loss of Pupae in Cold

Individuals that pupated from August 30 to September 6 were used for the experiments. They were acclimated at 20 °C on September 7 for a day and then at 15 °C on September 8 for a day. Individuals (pupae) of each dietary group were divided into two groups and exposed to either 3.0 (±0.01) or 7.4 (±0.06) °C under constant darkness in incubators (temperature in incubators was monitored every hour using loggers, ThermoManager, TOA Industry Co., Ltd., Tokyo, Japan). Relative humidity was 50 ± 10% at both temperature settings. According to the Meteorological Agency of Japan, the mean temperature of Kanazawa from January to March was 4.57 (±1.07 in SD) °C with a range of 1.8 to 7.3 °C (1970–2021). It is therefore considered that the experimental temperature of 3.0 approximately represents a condition of relatively cold winter and that of 7.4 °C performs a condition of warmer winter.

All pupae were weighed again 30 days after cold exposure. At that time, approximately one-third of pupae were dissected to determine survival. In this study, dissected pupae from which white hemolymph oozed were judged to have been alive because living diapausing pupae have usually been observed to have white hemolymph in our preliminary study. For pupae that were judged to be dead, fungal infection was further recorded; those from which non-white hemolymph or liquid at least partly oozed were judged to be infected. The pupae that were not subjected to dissection were subsequently kept at cold temperatures (3.0 or 7.4 °C) and weighed 60 days after exposure. Among these pupae, approximately half were collected and determined for survival and fungal infection. The remaining pupae were further kept at cold temperatures and examined for weight, survival, and fungal infection 90 days after cold exposure. Among a total of 162 pupae that were exposed to 7.4 °C, five individuals had almost completed metamorphosis in the pupal case in 90 days; they were categorized as “alive”. Pupae suspected to be infected by fungi and some dead pupae without any sign of fungal infection were retained in 99.5% ethanol for molecular analysis.

### 2.4. Fungal Identification

DNA was extracted from all pupae that were assumed to be killed by fungal infection and a few negative controls (dead pupa without any sign of fungal infection), and ITS (intertranscribed spacer sequence of the ribosomal RNA genes) fragment was amplified through polymerase chain reaction (PCR) using a primer pair ITS4/ITS5 [14]. PCR was performed with KOD FX taq DNA polymerase (Toyobo Co., Ltd., Osaka, Japan) according to the manual provided by the supplier. The amplified DNA products were electrophoresed and the target band of approximately 700 bp were cut out to extract DNA. Cycle sequencing reaction was performed on the isolated fungal DNA, and sequencing was conducted with an ABI Prism 3130xl Genetic Analyzer (ThermoFisher Scientific Co., Ltd., Waltham, MA, USA). The obtained sequences were BLAST-searched to identify fungal species and sequences showing 99% or higher coincidence were recorded.

### 2.5. Data Analysis

We used generalized additive model (GAM) with unity link function and normal distribution error to select the most influential environmental parameter on the abundance of overwintered population (N1). Using the number of overwintering populations as the objective variable, all combination patterns of explanatory variables and AICs were calculated. Finally, the model with the smallest AIC value was calculated as the best model. We applied logistic regression analyses to examine the effects of exposure temperature, duration of exposure, and larval food plants on the survival of pupae. The effects of exposure temperature on weight loss of living pupae were analyzed using analysis of covariance with duration of exposure and larval diet as covariables. The relationship between weight at pupation and survival was analyzed by logistic regression. Generalized additive model was performed using R v.3.6.1. (R Core Team, 2019). The other statistical analyses were done with JMP 11.2.1 (SAS Institute, Cary, NC, USA).

## 3. Results

### 3.1. Relationship between Fall Webworm Outbreaks and Meteorological Conditions

In total, 27,040 *Hy. cunea* males were collected in 18 years (mean ± SD = 1502 ± 943, with a range of 97 and 2834). As has previously been referred to, the moth catch showed two annual peaks (Figure 1). In this study, the effects of winter coldness and temperature conditions in spring on the abundance of adult individuals of the overwintered generation (i.e., the size of the first peak, N1) were analyzed by GAM. The results showed that the most influential factor was winter coldness (mean daily temperature of January to February, *p* = 0.043) (Table 1) and high temperatures in the previous June (*p* < 0.0001) (Table 1); the hotness of the previous June and winter warmness showed a negative relationship (Table 1). It has been reported in the past that summer heat inhibits the growth of *Hy. cunea*. However, winter warmth has never been reported to have a negative effect on populations. Figure 2 illustrates the correlation between N1 size and winter temperatures between 2004 and 2021. We tested if the temperature during winter diapause affects pupal mortality by laboratory experiments.

### 3.2. Survival, Weight Loss, and Fungal Infection of Pupae in Cold

Mortality of pupae increased when the duration of exposure to cold was prolonged (Table 2). When kept at 3.0 °C, 7 (4.0%) out of 176 individuals died in total, and 4 (57.1%) of dead pupae were infected with fungi. When kept at 7.4 °C, 27 (16.7%) out of 162 individuals died in total, and 26 (96.3%) of them were infected (Table 2). The logistic regression analyses revealed that exposure temperature, duration of exposure, and larval food plants significantly affected pupal mortality (Table 3). The interaction between the duration of exposure and larval food plants was significant (Table 3). This is because the mortality of sweetgum-fed individuals particularly increased when exposed to cold for 90 days in comparison with mulberry-fed individuals (Table 2). It is noteworthy that sweetgum is one of the major plants grown on the roadside in Kanazawa City and all larvae used in the present experiments were collected from sweetgum. 

Weight loss during diapause was analyzed excluding individuals that had died by the time of measurements (i.e., 30, 60, or 90 days after exposure to cold). The pupal weight relative to the initial weight (i.e., weight when pupae were exposed to cold) is shown in Table 4 and Figure 3 The results of the ANCOVA showed that pupal weight was significantly decreased when exposed to 7.4 °C than to 3 °C; when the duration of exposure was prolonged; and when they were fed with sweetgum leaves than with mulberry leaves (Table 4 and Table 5, Figure 3). The interaction of temperature and exposure duration was also significant (Table 5).

There was a strong negative relationship between the initial pupal weight and mortality; lighter pupae at the start of cold exposure showed significantly higher mortality than heavier pupae (Logistic regression, N = 338, χ^2^ = 13.1, *p* = 0.0003). 

### 3.3. Identification of Fungi

There was no amplified product in all negative controls (N = 8, dead pupa showing no sign of fungal infection) using primer pair ITS4/ITS5 whereas all 30 dead pupae showing signs of fungal infection amplified the target band with the primer pair. All nucleotide sequences amplified from 30 dead pupae showing signs of fungal infection showed 99% or higher matching with those from the following six species: *Penicillium citrinum*, *Penicillium gliseofulvem*, *Penicillium glabrum*, *Aspergillus niger*, *Aspergillus tubingensis*, and *Fusarium solani*. However, these three *Penicillium* species and the two *Aspergillus* species differ little in the sequence of the present ITS regions. For accurate species identification, a comparison of other DNA regions or an examination of sporocarp morphology is required.

## 4. Discussion

Although it is argued that insect population growth will be accelerated by global warming [1], the IPCC [15] generally predicts that the extinction rate of animals will accelerate 1000 times as much as before. The process of how warming affects the extinction of insects may vary. We studied the meteorological conditions and insect population growth in a city where 18-year records of fall webworm *Hyphantria cunea* population dynamics are preserved. It has been observed in the field study in Kanazawa that the abundance of *Hy. cunea* individuals in spring were negatively correlated with the winter temperature. In agreement with this field observation, mortality of diapausing pupae was higher when exposed to 7.4 °C than when exposed to 3.0 °C. It was also observed that almost all pupae that died at 7.4 °C showed signs of fungal infection. It is not determined in this study whether the fungal infection is a cause or result of mortality, i.e., there is a possibility that fungi proliferated after the pupae had died. Some *Aspergillus* species are ubiquitous, and silk moths up to the third instar are vulnerable to being infected with this fungus and dying [16]. The infectivity of wintering pupae of the fall webworm has not been investigated. Additional experimental work is needed to answer this question.

It is further confirmed in this study that pupae kept at 7.4 °C lost their weight more rapidly than those kept at 3.0 °C and that lighter pupae showed higher mortality when exposed to cold. The rapid weight loss at a higher temperature would be due to the metabolic rate being higher. We do not know how weight loss affects mortality in the moth species. Individuals that spend more energy on metabolism and lighter individuals with fewer energy resources may not have enough resources for immunity and therefore may be vulnerable to infection by fungi or microorganisms. Further study is also needed on these subjects.

This phytophagous pest is no longer considered a severe problem in many regions in lower latitudes. However, there are no statistics by governments or authorities, while it has now become prevalent at higher latitudes in Japan [17]. This change could be partly attributed to warmer winters due to global warming. After the invasion and rapid expansion into Japan, the distributional change in this species is not an expansion but rather a migration, with a reduction in distribution southwards. This study suggests that winter temperatures are critical for the survival of this pest and that increased winter temperatures make low altitudes too warm for this pest to survive. Overall, our experiments suggest that global warming is likely to adversely affect overwintering pupa through weight loss and fungal infections in mild winters. Looking back on the expansion of the distribution of the moth in the Japanese islands, it expanded its distribution to parts of the Shikoku and Kyushu islands in the 1970s but did not expand southward afterward [5]. It implies that the limit of low-latitude distribution originally existed, and the southern limit may be moving northward due to global warming. Since the genetic background of this species distributed in Japan has the minimum variations, the single invasion is estimated to spread all over Japan [5]. Therefore, if global warming continues as it is, this moth species is expected to expand its north limit but disappear from the southern limit area simultaneously. The current state of local population reduction due to the global warming of the moth, shown in this study, would caution us about the fate of temperate insect species with the winter diapause stage.

## Figures and Tables

**Figure 1 insects-14-00534-f001:**
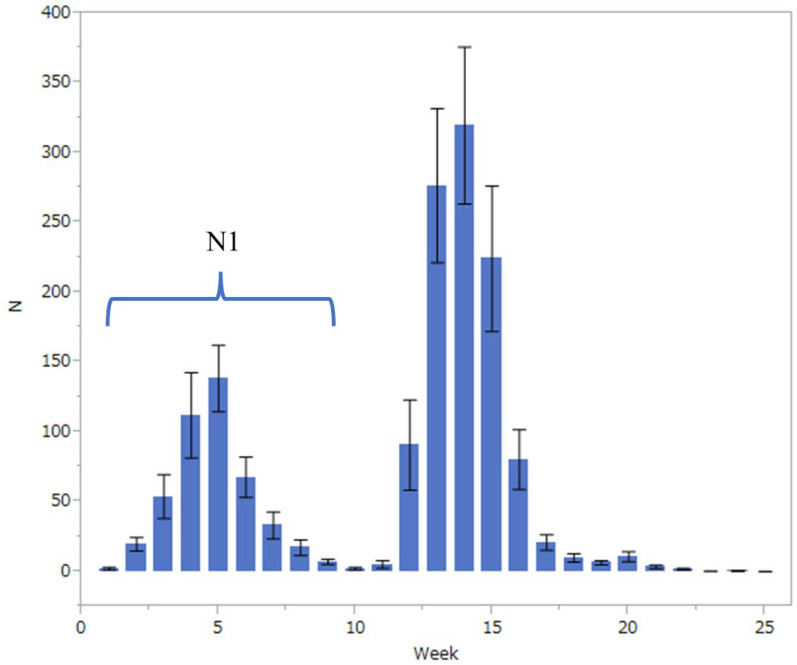
Number (mean ± SD) of weekly captures of *Hyphantria cunea* males in 25 weeks from late April to early October for 18 years from 2004 to 2021 in Kanazawa City.

**Figure 2 insects-14-00534-f002:**
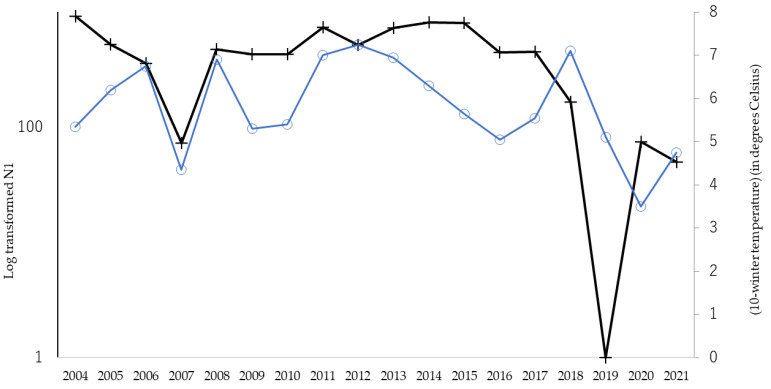
Yearly fluctuations in abundance of N1 (overwinter generation) and modified winter temperature between 2004 and 2021. Dark line with cross marks shows the log-transformed number of overwintering males (the vertical axis on the left), and blue line with round marks shows winter temperature as (10-mean temperature of Jan. to Feb.) (°C) (the vertical axis on the right).

**Figure 3 insects-14-00534-f003:**
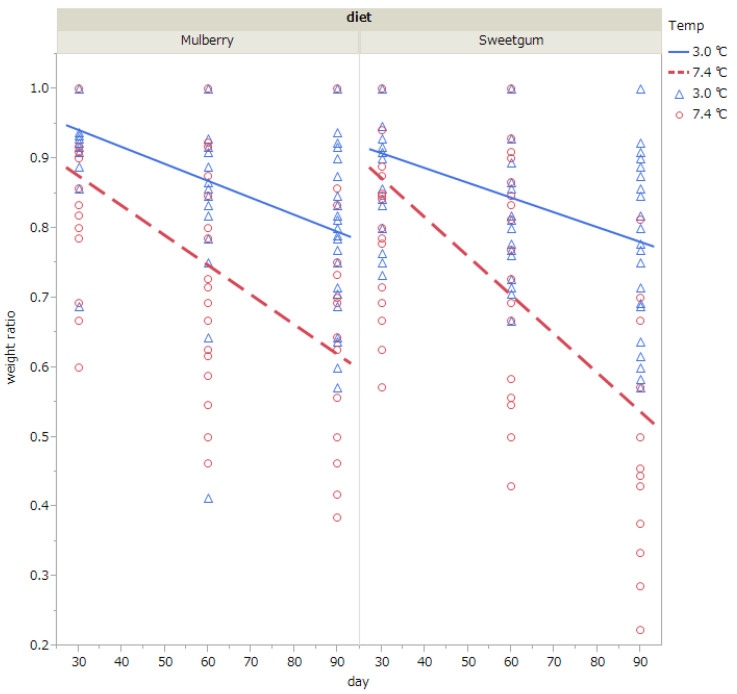
Changes in body weight relative to the initial weight (i.e., weight when pupae were exposed to cold) when diapausing pupae of *Hyphantria cunea* were exposed to 3.0 °C (blue triangle) or 7.4 °C (red circle) for 30, 60, or 90 days. Pupae were raised on mulberry or sweetgum leaves at the larval stage. Pupae that died by the time of measurement (i.e., 30, 60, or 90 days after exposure to cold) were excluded from the analysis. The straight lines show regression lines (blue line, 3.0 °C)(red dotted line, 7.4 °C). Number of pupae subjected to measurement were 68 (sweetgum, 7.4 °C), 90 (sweetgum, 3.0 °C), 64 (mulberry, 7.4 °C), and 76 (mulberry, 3.0 °C).

**Table 1 insects-14-00534-t001:** Summary of GAM to explain N1{male moth catches between 1st and 10th week/(April 21 to June 29) in 2004–2021. The best model was selected by AIC value. Formula: N.1 ~JanFebT + Jun.h.last + Jun T + May T + NovMarT, Family; Gaussian, Link function: identity, R-sq. (adj) = 0.197, Deviance explained = 22.9%, GCV = 2185.2, Scale est. = 2083.9, AIC = 1591.

		Parameter Coefficients			
Parameter	Definition	Estimate	Std. Error	t-Value	*p*
(Intercept)		157.91	55.05	2.87	0.005
Jan–Feb T	Average daily temperature of Jan. to Feb.	−18.82	9.21	19.58	0.043
Jun-highT-last	Cumulative value of daily average temperature above 25 ° C minus 25 ° C in June of previous year	−3.13	0.70	−4.45	<0.0001
May T	Cumulative value of daily average temperature above 10 ° C minus 10 ° C in May	0.15	0.13	1.11	0.268
Jun T	Cumulative value of daily average temperature above 10 ° C minus 10 ° C in June	0.04	0.05	0.83	0.409
NovMarT	Average daily temperature from Nov. to March.	−9.68	10.98	−0.88	0.380

**Table 2 insects-14-00534-t002:** Number of pupae that died (D) and were infected by fungi (F) after exposure to 7.4 or 3.0 °C for 30, 60, or 90 days. They were raised with sweetgum or mulberry at the larval stage.

Duration of Exposure		30			60			90			Total		
Temp	Diet	N *	D	F **	N	D	F **	N	D	F **	N	D	F **
7.4 °C	Sweetgum	26	0	0	28	2	2	30	14	14	84	16	16
	Mulberry	30	2	2	26	3	2	22	6	6	78	11	10
3.0 °C	Sweetgum	26	0	0	22	0	0	46	4	2	94	4	2
	Mulberry	28	0	0	23	0	0	31	3	2	82	3	2

* Number of pupae dissected. ** Infection by fungi was determined by PCR amplification.

**Table 3 insects-14-00534-t003:** Summary of logistic regression analysis on pupal mortality. Larvae were raised with two plant species (mulberry or sweetgum) and pupae were exposed to 3.0 or 7.4 °C for 30, 60, or 90 days.

	χ^2^		*p*
	**46.47**		**<0.0001**

**Exposure temperature (T)**	**12.14**		**0.0005**
**Duration of exposure (D)**	**16.06**		**<0.0001**
**Food plant (F)**	**6.98**		**0.008**

T × D	0.24		0.621
T × F	1.03		0.311
**D × F**	**14.5**		**0.0001**

**Table 4 insects-14-00534-t004:** Changes (mean and SD) in pupal weight relative to the initial weight (weight when pupae were exposed to cold) after exposure to 3.0 or 7.4 °C for 30, 60, or 90 days. They were raised with sweetgum or mulberry in the larval stage. Pupae that died by the time of measurement (30, 60, or 90 days after exposure to cold) were excluded from measurement.

	3.0 °C						7.4 °C					
	Sweetgum			Mulberry			Sweetgum			Mulberry		
Duration of Exposure	N	Mean	SD	N	Mean	SD	N	Mean	SD	N	Mean	SD
30	25	0.913	0.085	25	0.949	0.070	26	0.840	0.121	25	0.882	0.124
60	23	0.834	0.092	23	0.851	0.124	26	0.766	0.169	23	0.733	0.150
90	42	0.783	0.112	28	0.802	0.108	16	0.485	0.163	16	0.629	0.174

**Table 5 insects-14-00534-t005:** Summary of ANCOVA on pupal weight relative to the initial weight (weight when larvae were exposed to cold).

Whole Model: d.f. = 6, F-Value = 31.46, *p* < 0.0001.		
Factors	t-Value	*p*
**Exposure temperature (T)**	**−8.6**	**<0.0001**
**Duration of exposure (D)**	**−11**	**<0.0001**
**Food plant (F)**	**−2**	**0.0441**
T × F	−0.57	0.5723
**D × T**	**−4.3**	**<0.0001**
F × D	−0.58	0.5641

## Data Availability

The data presented in this study are available on request from the corresponding author. The original entomological data is owned by Kanazawa City hall.

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
