# Peer review of "Mild Winter Causes Increased Mortality in the Fall Webworm *Hyphantria cunea* (Lepidoptera: Arctiidae)"

_insects, 2023, doi:10.3390/insects14060534_

Round 1

Reviewer 1 Report

insects-2374951

In this research, the authors indicate that increases in winter temperature might be associated with an invasive pest decline in Japan. Based on field data and laboratory experiments, they found that mild winter temperatures cause pest mortality by reducing the weight of pupae interacting with plant hosts. Although the laboratory supports pupal weight as a potential mortality factor, the field data is unclear regarding insect decline.

The authors used a GLM to analyze the effects of month averaged temperatures on pest catch. However, in doing that, they should have noticed the impact of time in the analysis. Thus, it is necessary to analyze this dataset considering time as a factor. For instance, the data could be interpreted as a time series since the authors collected an extended dataset in the same time frame (e.g., GAM models). Figure 1 did not show the number of individuals per year; the authors accumulated the number of moths according to the week of the year. In addition, no weather data shows an increase in winter temperatures. They only mention average measures in lines 142-144. Plus, only temperature was tested in this study, so climatic factors are a broad term. Did the authors expect another climatic factor to affect this pest population dynamic?

This study could be accepted for publication, but a time series analysis needs to take place first.

L 11: high altitudes may provide a better picture of pest movement.

L 11-12: Is Kanazawa a high-altitude city?

L 17-18: This may invalidate the results since the insect died due to pathogens infection, not temperature conditions. May the authors consider removing it or specify the percentual of dead individuals?

L 69-80: The study’s objective is not presented as the authors’ primary presents the methodology and partial results.

L 98: The authors only tested temperature. Climatic factors are a broad term for that.

L 175-182: How do the authors account for temporal autocorrelation in the data since they were repeatedly measured at the exact location over time? It also implies pseudoreplication, which needs to be indicated in how they deal with it. Are the predictors correlated? This needs to be shown to avoid the multicollinearity effect in the model presented.

NA

Author Response

Reply to the Reviewer 1

We thank you for your valuable and constructive comments.

We prepared our responses to the reviewers' comments as below.

Our reply is shown in blue.

Comments and Suggestions for Authors insects-2374951

In this research, the authors indicate that increases in winter temperature might be associated with an invasive pest decline in Japan. Based on field data and laboratory experiments, they found that mild winter temperatures cause pest mortality by reducing the weight of pupae interacting with plant hosts. Although the laboratory supports pupal weight as a potential mortality factor, the field data is unclear regarding insect decline. The authors used a GLM to analyze the effects of month averaged temperatures on pest catch. However, in doing that, they should have noticed the impact of time in the analysis. Thus, it is necessary to analyze this dataset considering time as a factor. For instance, the data could be interpreted as a time series since the authors collected an extended dataset in the same time frame (e.g., GAM models).

Reply. Thank you for providing us with valuable input. However, in this study, we applied only meteorological parameters to evaluate if simple environmental parameters can predict moth population if they increase or decrease. The study was not financially funded by Kanazawa city halls but our primary purpose was to find a simple environmental predictor for moth abundance in every year.

Our entomological data is collected weekly by Kanazawa City using 18 traps in total at nine locations to track male moth catches. The meteorological data is downloaded from Kanazawa MET that is the closest MET to all sampling locations. Microclimate data was also collected for each trap installation site, but those data only showed parallel temperature changes with the data from the Kanazawa MET therefore we did not continue to record the microclimatic data.

As the reviewer is anxious about pseudo-replication, we wish to find similar data at remote places with different temperatures from Kanazawa. But we could not find any other local government where they have enough data on the moth occurrence like Kanazawa city because Kanazawa city hall has started its sampling with the advice of our senior staff. We appreciate the reviewer's suggestion about time-series data analysis and have consulted with statistics experts. The local populations collected at the nine locations will not be freely mixed but are strongly influenced by the previous generation's population at the same location.

When we have added the number of individuals from the previous generations as explanatory variables for the number of individuals in the overwintering generation, previous generation abundance was a significant parameter next to the winter coldness. The winter coldness was much more significant in explaining the size of N1 than the moth abundance in previous generations. This parameter represents the uniqueness of the collection site. As meteorological data remains the same for all collection sites and is not unique to any of them, examining changes over time in each population using the number of individuals in the previous generation as an explanatory variable is necessary. This is because weather information across locations is identical, and changes over time do not represent differences between locations.

Figure 1 did not show the number of individuals per year; the authors accumulated the number of moths according to the week of the year. In addition, no weather data shows an increase in winter temperatures. They only mention average measures in lines 142- 144. Plus, only temperature was tested in this study, so climatic factors are a broad term. Did the authors expect another climatic factor to affect this pest population dynamic? This study could be accepted for publication, but a time series analysis needs to take place first.

Reply. Our research team has prepared two comprehensive figures to aid in understanding the complexities of overwintering generations and the fluctuations in the size of N1 and winter temperatures between 2004-2019. Figure 1 provides a specific definition of N1 size, while Figure 2 illustrates the correlation between N1 size and winter temperatures. It should be noted that our data for 2020 and 2021 is not included in Figure 2 due to changes in pheromone trap models and sampling locations. However, we provide detailed explanations for these changes in the accompanying text.

L 11: high altitudes may provide a better picture of pest movement.

Reply. Thank you for your valuable feedback. If Reviewer 1 recommends conducting the observation at higher altitudes, we have diligently searched for relevant and reliable data on similar pest observations from local government sources in mountainous areas, but unfortunately, most of them only possess records of resident requests pertaining to pest-related issues. Thank you for the comment.

L 11-12: Is Kanazawa a high-altitude city?

Reply. Thank you for the comment. Kanazawa faces the sea and is not located at a higher altitude. All sampling locations are within 15-40m asl as in L 89. We added the information in the text.

L 17-18: This may invalidate the results since the insect died due to pathogens infection, not temperature conditions. May the authors consider removing it or specify the percentual of dead individuals?

Reply. Thank you for sharing your insights. We must take into account that both high and low temperatures have been found to have adverse effects on pupal survival. Numerous studies have examined the direct impacts of extreme temperatures on pupal mortality. However, the winters in Kanazawa are not severe enough to cause such mortality. Our current focus is on investigating the long-term temperature conditions that overwintering pupae are exposed to, and how they affect various biological factors. The metamorphosis undergone by moths during the pupal stage makes it challenging to determine their viability. If a pupa perishes, it typically mummifies quickly, while surviving pupae emerge in the spring, allowing for a viability assessment. However, determining the cause of death at the time of occurrence remains a challenge.

L 69-80: The study’s objective is not presented as the authors’ primary presents the methodology and partial results.

Reply. Thank you for the comment. We rephrased the relevant part (L68-81) to show the study objective. Our initial goal was to discover a straightforward way to forecast moth population for the upcoming year using meteorological data.

L 98: The authors only tested temperature. Climatic factors are a broad term for that.

Reply. Thank you for providing feedback on our research. We conducted a preliminary study to investigate various factors, including temperature fluctuations and snowfall patterns, that influence the population of overwintering moths. During the initial decade of our study (2004-2009), our analysis revealed a positive correlation between the scarcity of snowfall and the abundance of overwintering moths. However, we faced challenges in distinguishing the impact of winter temperature from snowfall during this period. With the accumulation of additional data after 2009, we observed that winter temperatures are a more reliable predictor of overwintering moth abundance, independent of snowfall levels. Based on our findings, we conclude that winter temperatures play a significant role in determining the population of overwintering moths.

L 175-182: How do the authors account for temporal autocorrelation in the data since they were repeatedly measured at the exact location over time? It also implies pseudoreplication, which needs to be indicated in how they deal with it. Are the predictors correlated? This needs to be shown to avoid the multicollinearity effect in the model presented.

Reply. We appreciate your comment. The reviewer expressed concerns about pseudo-replication, so we attempted to locate comparable data from other locations with varying temperatures outside of Kanazawa. Unfortunately, we were unable to find any other local governments with sufficient data on moth occurrences similar to Kanazawa city.

Reviewer 2 Report

The present study entitled “Mild winter causes increased mortality in the Fall webworm Hyphantria cunea (Lepidoptera: Arctiidae)” investigates the effects of changing (i.e. warming) winter climate on survival in the Fall webworm (Hyphantria cunea). The authors analyzed an 18 years long record of occurrence of Hy. cunea in relation to winter temperatures. The authors found a negative correlation between spring emergence and winter temperatures. A subsequent laboratory experiment confirmed the negative relationship between temperature and survival. In addition, the laboratory experiment revealed a possible effect of fungal infection on mortality at higher temperatures.

The present manuscript is well written and I have no serious reservations about the methods and interpretation of the results. Some aspects of the methods could have been done "better" or differently, but on the other hand, changes could have made the results more difficult to interpret. For example, it would be better to assess survival as adult emergence (at least at the end of the experiment, when diapause was likely terminated), however it could make the other analyses impossible or difficult. Considering the main purpose of the study, it seems unnecessary to me to include the influence of diet in the design of the experiment, although the results show its clear effect on some parameters. It could be interesting (but likely problematic) to analyze weight also in dead pupae. The number of references is rather low for the article type, however to my opinion, the number is sufficient for the present study and the references seem relevant. Although the language is very good, the text could benefit from a little editing.

To my opinion, the present manuscript is of good quality and suitable for publication in Insects after a minor revision.

Specific comments:

L40, L253: Change “ICPP” to IPCC.

L278-281: Confusingly written. Although the sentence is comprehensible after some effort, it deserves editing to improve clarity.

The language is very good, however the text could benefit from a little editing.

Author Response

Reply to reviewer 2

We thank you for your careful reading and encouraging comments on the manuscript.

Our reply is written in blue.

Comments and Suggestions for Authors

The present study entitled “Mild winter causes increased mortality in the Fall webworm Hyphantria cunea (Lepidoptera: Arctiidae)” investigates the effects of changing (i.e. warming) winter climate on survival in the Fall webworm (Hyphantria cunea). The authors analyzed an 18 years long record of occurrence of Hy. cunea in relation to winter temperatures. The authors found a negative correlation between spring emergence and winter temperatures. A subsequent laboratory experiment confirmed the negative relationship between temperature and survival. In addition, the laboratory experiment revealed a possible effect of fungal infection on mortality at higher temperatures.

The present manuscript is well written and I have no serious reservations about the methods and interpretation of the results. Some aspects of the methods could have been done "better" or differently, but on the other hand, changes could have made the results more difficult to interpret. For example, it would be better to assess survival as adult emergence (at least at the end of the experiment, when diapause was likely terminated), however it could make the other analyses impossible or difficult. Considering the main purpose of the study, it seems unnecessary to me to include the influence of diet in the design of the experiment, although the results show its clear effect on some parameters. It could be interesting (but likely problematic) to analyze weight also in dead pupae. The number of references is rather low for the article type, however to my opinion, the number is sufficient for the present study and the references seem relevant. Although the language is very good, the text could benefit from a little editing.

To my opinion, the present manuscript is of good quality and suitable for publication in Insects after a minor revision.

Reply. We deeply thank the encouragement.

Specific comments:

L40, L253: Change “ICPP” to IPCC.

Reply. We corrected the mistakes.

L278-281: Confusingly written. Although the sentence is comprehensible after some effort, it deserves editing to improve clarity.

Reply. We rephrase the part as,,

Our study suggests that increased low-latitude winter temperatures are critical for this pest and that it is too warm to survive at low latitudes. Overall, our experiments suggest that global warming is likely to adversely affect overwintering pupa through weight loss and fungal infections in mild winters.

Comments on the Quality of English Language

The language is very good, however the text could benefit from a little editing.

Reply. We checked the whole text again.

We appreciate your valuable time and comments.

Round 2

Reviewer 1 Report

#insects-2374951

In this new version, the authors presented subtle changes in the manuscript. However, it is unclear that the species studied is declining (Figure 2, black line). Interestingly, it seems that this species decline is a multifactorial event, including parasitism, predation, and pathogen factors (lines 60-70). Nevertheless, the authors mainly attributed to climate change the event observed. Some key points need to be clarified yet. For instance, how do the authors deal with pseudoreplication in the models regarding field data collection? Did they sum the values? Thus, the manuscript needs to be revised before publication.

Title: May the authors consider changing it, as mild winter is not the only cause of population decline. Please see lines 69-71 and the authors' report regarding fungi infection.

 L 17 and 32: This sentence may lead to readers' misinterpretation as the authors mentioned temperature as the cause of mortality, also attributed to fungi infection. The insect decline could be multifactorial, not only related to the temperature.

 Fig. 2: This figure did not indicate that the population is in decline, as mainly claimed by the authors. There is a reduction only in 2019.

Author Response

Reply to the reviewer        

We thank you for your careful reading and encouraging comments on the manuscript.

Our reply is written in blue.

We agree with Reviewer 1 that there are multiple reasons for the population decline. We are currently working on another manuscript that will provide further explanation on this matter. The focus of this manuscript will be on the winter diapausing generation.

In this new version, the authors presented subtle changes in the manuscript. However, it is unclear that the species studied is declining (Figure 2, black line). Interestingly, it seems that this species decline is a multifactorial event, including parasitism, predation, and pathogen factors (lines 60-70). Nevertheless, the authors mainly attributed to climate change the event observed. Some key points need to be clarified yet. For instance, how do the authors deal with pseudoreplication in the models regarding field data collection? Did they sum the values? Thus, the manuscript needs to be revised before publication.

Our reply.

In the third version, we changed the way of analysis as the reviewer has suggested.

We applied generalized additive model (GAM) this time.

Title: May the authors consider changing it, as mild winter is not the only cause of population decline. Please see lines 69-71 and the authors' report regarding fungi infection.

Our reply.

It will be true though mild winter was chosen as an influential climatic factor by our analysis.

And we do not have any other field data except for the male moth number.

 L 17 and 32: This sentence may lead to readers' misinterpretation as the authors mentioned temperature as the cause of mortality, also attributed to fungi infection. The insect decline could be multifactorial, not only related to the temperature.

Our reply.

It is true. We rewrote the part as

However, the impact of warm winters on populations in the field can be more complicated and multifaceted.

 Fig. 2: This figure did not indicate that the population is in decline, as mainly claimed by the authors. There is a reduction only in 2019.

Our reply.

We have updated Figure 2 with data reflecting the population decline until 2021. The recorded annual moth catches for the years 2019, 2020, 2021, and 2022 were 170, 97, 97, and 55, respectively. The average annual moth catches from 2004 to 2018 was 1778.33. However, locating larval nests within the town has become increasingly challenging.

We would like to express our sincere gratitude to the reviewers for their diligent review and valuable feedback.
